# Segmented Two-Dimensional Progressive Polynomial Calibration Method for Nonlinear Sensors

**DOI:** 10.3390/s24217058

**Published:** 2024-11-01

**Authors:** Jae-Lim Lee, Dong-Sun Kim

**Affiliations:** Department of Semiconductor Systems Engineering, Sejong University, Seoul 05006, Republic of Korea; jaelim07@sju.ac.kr

**Keywords:** sensor calibration, segmented calibration, 2D calibration

## Abstract

Nonlinearity in sensor measurements reduces the sensor’s accuracy. Therefore, accurate calibration is necessary for reliable sensor operation. This study proposes a segmented calibration method that divides the input range into multiple sections and calculates the optimized calibration functions for each one. This approach reduces the overall error rate and improves the calibration accuracy by isolating distinctive regions. The modified progressive polynomial calibration technique is used to calculate the calibration function. This algorithm addresses the computational complexity, allowing for reduced polynomial degrees and improving the accuracy. The segmented calibration method achieves a significantly lower error rate of 0.000006% compared to the original single calibration method, which has an error rate of 0.0823%, when using the same six calibration points and a fifth-degree polynomial function. This method maintains improved accuracy with fewer calibration points, and its ability to reduce the computational complexity and calculation time while using lower polynomial degrees is confirmed. Additionally, it can be extended to two dimensions to reduce the errors caused by cross-sensitivity. The results from a two-dimensional simulation show a reduction in the error rate ranging from 15.84% to 2.07% in an 8-bit signed fixed-point system. These results indicate that the segmented calibration method is an effective and scalable solution for various typical sensors.

## 1. Introduction

In recent years, sensors have been used in various fields, such as medicine, healthcare, agriculture, automotives, and IoT [1,2,3,4,5,6]. The sensors’ accuracy is critical in ensuring high performance and reliability in these fields [7,8,9]. However, sensors exhibit nonlinearity due to environmental conditions such as the temperature and humidity or the equipment’s inherent characteristics. For example, pressure sensors and relative humidity sensors are influenced by the temperature, resulting in different output values due to temperature variations. The effect of the temperature on the output of a relative humidity sensor can be seen in Figure 1. Figure 1 shows the relative humidity on the x-axis, and the capacitance, which is the sensor’s output in response to humidity, on the y-axis. The graph illustrates the changes in capacitance across seven different temperature conditions, ranging from T = 10 °C to 70 °C in 10 °C increments. It indicates that the capacitance values vary with the temperature, confirming that the output of the relative humidity sensor is affected by the temperature. In addition to environmental factors, the boundary conditions of the sensor’s measurement range often exhibit rapid changes or significantly higher error rates compared to the central range. This is especially evident near the upper and lower limits of the range [10,11], where nonlinear errors become more pronounced and sensor saturation tends to occur. These factors suggest that additional calibration may be necessary to ensure accurate measurements. Therefore, it is also necessary to perform additional calibration, not only for the environmental conditions but also for the boundary regions of the sensor’s measurement range, to enhance the overall accuracy.

Many calibration methods have been developed to address these inaccuracies [12]. The most commonly used methods are curve fitting [13,14,15,16,17,18], lookup tables [19,20], and artificial neural networks (ANNs) [21,22,23,24,25]. The lookup table method is the simplest, but its accuracy is limited and it occupies a large amount of memory. The ANN method can be used to calibrate the sensor output characteristics through repeated learning and training. However, it has disadvantages, such as slow convergence, susceptibility to local extremes, and uncertain weight thresholds, making the generalization capacity of the neural network highly dependent on these conditions. By comparison, the curve fitting method balances simplicity and flexibility, offering faster computation with smaller memory requirements while maintaining the ability to handle various data patterns and environmental conditions. The most commonly used type of curve fitting is the polynomial approach. In this study, a type of curve fitting known as progressive polynomial calibration (PPC) is applied.

This work proposes a method that divides the sensor’s range into segments and applies an optimized calibration function to each segment. This approach reduces the errors that arise at the boundaries of the sensor’s range, improving the overall accuracy. Additionally, by extending the polynomial calibration function to two dimensions, the method compensates for output variations caused by the environmental conditions. This scalable calibration algorithm can be applied to a wide range of sensors, reducing the number of errors due to environmental factors and enhancing the overall accuracy of the sensor system. Section 2.1 explains the proposed segmented calibration method. This is followed by Section 2.2, which introduces the modified progressive polynomial calibration algorithm that is used alongside the segmented method. Section 3 presents the MATLAB simulations, in which we apply this method for sensor calibration. Specifically, Section 3.1.1 covers a one-dimensional calibration simulation using the same input for comparison with other studies, while Section 3.1.2 focuses on one-dimensional simulations using data from a pressure sensor. Section 3.2 extends the simulation to two dimensions by adding the effect of the temperature to the pressure data used in Section 3.1.2. Finally, Section 4 provides the conclusions of this work.

## 2. Proposed Method

### 2.1. Segment Calibration Method

The segmented calibration technique divides the sensor’s input range into multiple sections, applying a calibration function to each segment. Rather than using a single calibration function across the entire input range, this method allows for more precise calibrations in areas where the nonlinearities are more significant, particularly near the extremes of the sensor’s range. The proposed method is described in the following.

1.Determine the sections and boundaries: Specify the number of sections and boundary values according to the response characteristics of each sensor. In this study, considering the characteristics of the simulation sensor, the range is divided into three sections where the linear characteristic changes significantly. The boundary sections are defined sequentially as the low, mid, and high sections. The boundary between the low section and mid section is denoted as $B_{1}$, and the boundary between the mid section and high section is denoted as $B_{2}$.2.Split the input data: The input data required for sensor calibration, i.e., the physically measured data from the sensor and the reference data, are divided according to the respective sections. To prevent data discontinuities and to ensure smooth transitions, the end values of each section overlap. In particular, the last value of the low section is the same as the first value of the mid section, and the last value of the mid section is the same as the first value of the high section.3.Calculate the calibration function: The calibration functions corresponding to each section are calculated individually. A progressive polynomial calibration algorithm, a type of curve fitting method, is used for this purpose. The calibration function for the low section is denoted as $f_{l}$, that for the mid section as $f_{m}$, and that for the high section as $f_{h}$.

Applying a transfer function customized to each segment minimizes the errors, improving the sensor’s overall accuracy. As shown in Figure 2, the single method has only one calibration function (f(x)), whereas the segmented method has calibration functions for each section ($f_{l}\left(x\right)$, $f_{m}\left(x\right)$, $f_{h}\left(x\right)$). Moreover, each calibration function is continuous. This segmentation enables more accurate corrections in the highly nonlinear regions near the sensor’s upper and lower limits.

### 2.2. Progressive Polynomial Calibration Algorithm

#### 2.2.1. One-Dimensional Modified Progressive Polynomial Calibration (1D M-PPC)

PPC is a calibration method that involves gradually adding polynomials to compute the calibration function and coefficients. It keeps the previous correction values unchanged throughout the process. It ensures that the offset and gain correction values from step 1 and step 2 remain unchanged as further steps are implemented. Maintaining the alignment of the initial and final values with the reference values guarantees consistency, making PPC ideal for segmented calibration methods where consistency in the boundary values across the segments is crucial. The calibration points are generally selected in the following sequence: the first point is placed at one end of the sensor’s operating range, corresponding to offset calibration; the second point is placed at the other end of the range, corresponding to gain calibration; and additional calibration points are placed between the previously selected points, corresponding to linearity correction [26]. However, one limitation of PPC is that the polynomial degree increases exponentially as the number of steps increases, leading to greater computational complexity and longer processing times. Additionally, increasing the polynomial degree can cause the Runge phenomenon [24,27], resulting in inaccuracies.

This study implements the modified progressive polynomial calibration (M-PPC) [28] technique to address this issue, modifying it to overcome the limitations of PPC. M-PPC is developed so that the polynomial degree increases linearly with each step, thereby reducing the computational complexity and processing time while maintaining the calibration accuracy. The M-PPC method adjusts the calibration terms by directly reflecting the differences between the data points within each segment, and it adds correction terms based on $x-x_{i}$, allowing for the more effective control of the polynomial degree.

The transfer function for the 1D PPC algorithm is as follows:(1)$$f_{n}\left(x\right)=f_{n-1}\left(x\right)+a_{n}\cdot \prod_{i=1}^{n-1}\left(f_{i}\left(x\right)-y_{i}\right)$$

The coefficient for the 1D PPC algorithm is given by
(2)$$a_{n}=\frac{y_{n}-f_{n-1}\left(x_{n}\right)}{\prod_{i=1}^{n-1}\left(f_{i}\left(x_{n}\right)-y_{i}\right)}$$

The transfer function for the 1D M-PPC algorithm is as follows:(3)$$f_{n}\left(x\right)=f_{n-1}\left(x\right)+f_{n}^{\prime }\left(x\right)$$
(4)$$f_{n}^{\prime }\left(x\right)=b_{n}\cdot \prod {i=1}^{n-1}\left(x-x_{i}\right)$$

The coefficient for the 1D M-PPC algorithm is given by
(5)$$b_{n}=\frac{y_{n}-f_{n-1}\left(x_{n}\right)}{\prod_{i=1}^{n-1}\left(x_{n}-x_{i}\right)}$$

In the process of calculating the coefficients, the $y_{n}-f_{n-1}\left(x_{n}\right)$ term of $a_{n}$ and the $y_{n}-f_{n-1}\left(x_{n}\right)$ term of $b_{n}$ ensure that the values of the coefficients match the correction point. This term helps to ensure that the values at the boundary points are consistent. Compared to the conventional PPC method, where the degree increases exponentially as $2^{\left(n-1\right)}$ at the n-th step, the M-PPC method exhibits a more controlled increase, with the degree rising linearly by n−1. This improved calibration technique is optimized for section-by-section calibration and has demonstrated high accuracy and efficiency in experiments. The descriptions of the parameters used in the calculation are listed in Table 1.

#### 2.2.2. Two-Dimensional Modified Progressive Polynomial Calibration (2D M-PPC)

One-dimensional calibration can be extended to two dimensions due to the repetitive pattern of PPC. In two-dimensional calibration, the variable *z* is added to correct the cross-sensitivity caused by the environmental conditions. This extends the concept of fixing a single point in one dimension to an entire line, applying the same principle as in one-dimensional calibration. The original sensor variable *x* is calibrated a total of *N* times; simultaneously, the environmental condition variable *z* is calibrated a total of *M* times. The calibration order proceeds in the same sequence as in the one-dimensional calibration method, with *x* and *z* each following the order of offset, gain, and linearity correction. In the calibration process, the current calibration steps are denoted as *n* and *m*. With *n* fixed, the calibration is performed for *m* ranging from 1 to its maximum, *M*, and this process is repeated for each value of *n* from 1 to its maximum, *N*. Specifically, starting with n=1, the calibration proceeds through m=1,m=2,…,m=M. Then, for n=2, it is repeated from m=1 to m=M, continuing this pattern until n=N. Finally, a total of n=N and m=M calibrations are performed, resulting in a total of N×M calibrations. The calibration function can be calculated step by step using the following equations.

The transfer function for the 2D PPC algorithm is as follows:(6)$$f_{nm}\left(x,z\right)=f_{n,m-1}\left(x,z\right)+a_{nm}\prod_{j=1}^{m-1}\left(z^{\prime }-z^{\prime }j\right)\prod {i=1}^{n-1}\left(f_{im}\left(x,z\right)-y_{i}\right)$$

The coefficient for the 2D PPC algorithm is given by
(7)$$a_{nm}=\frac{y_{n}-f_{n,m-1}\left(x_{n},z_{m}\right)}{\prod_{i=1}^{n-1}\left(f_{iM}\left(x_{i},z_{M}\right)-y_{i}\right)\cdot \prod_{j=1}^{m-1}\left(z_{m}^{\prime }-z_{j}^{\prime }\right)}$$

The transfer function for the 2D M-PPC algorithm is as follows:(8)$$f_{nm}\left(x,z\right)=f_{n,m-1}\left(x,z\right)+f^{\prime }nm\left(x,z\right)$$
(9)$$f^{\prime }nm\left(x,z\right)=b_{nm}\cdot \prod_{i=1}^{n-1}\left(x-x_{i}\right)\cdot \prod_{j=1}^{m-1}\left(z-z_{j}\right)$$

The coefficient for the 2D M-PPC algorithm is given by
(10)$$b_{nm}=\frac{y_{n}-f_{n,m-1}\left(x_{n},z_{m}\right)}{\prod_{i=1}^{n-1}\left(x_{n}-x_{i}\right)\cdot \prod_{j=1}^{m-1}\left(z_{m}-z_{j}\right)}$$

In two-dimensional calibration, each time that a calibration point is added for cross-sensitivity, the number of calibration points increases by *N*. When applied to PPC, this significantly increases the order, leading to greater complexity and longer computation times. In PPC, each time that *M* increases by one point, the order increases by $2^{N-1}$. Therefore, the total calibration order becomes $2^{M(N-1)}$. However, when using the modified methodology, it becomes N×M−1, effectively reducing the order and decreasing the complexity and computational time for fast two-dimensional calibration. By applying this modified PPC to the segmented calibration method, the calibration time for multiple sections can be reduced, and the complexity can be lowered. The descriptions of the parameters used in the calculation are listed in Table 2.

## 3. Experiments and Results

The simulation was performed for three cases: (1) a one-dimensional calibration simulation for the correction of sensor data with nonlinear characteristics; (2) a one-dimensional calibration simulation for the correction of sensor data with linear characteristics; and (3) a two-dimensional calibration simulation. Each simulation was performed using Each simulation was performed using MATLAB version 9.14.0.2337262 (R2023a, Update 5), developed by MathWorks, Natick, MA, USA.

### 3.1. One-Dimensional Calibration Simulation

#### 3.1.1. One-Dimensional Calibration Simulation—A Nonlinear Sensor

This section considers the calibration of a sensor with nonlinear characteristics, comparing the segmented method to the single calibration method. During the simulation, the number of calibration points (CP) was varied among three, four, five, and six, and the results were compared with those from the single calibration method. The input range was normalized to 0 to 100.

The sensor’s characteristic is defined by
(11)$$f\left(x\right)=1-\exp\left(\frac{-x}{20}\right)$$

The results for the simulation with the number of CP set to six are presented in Table 3. The results were compared with those from a previous study [28]. PPC (single) and M-PPC (single) are general methods for single calibration, whereas M-PPC (segmented) is a method that divides the calibration into sections. In this simulation, the calibration was divided into three sections where the linearity changed drastically (Figure 3). As a result, the segmented method showed the lowest error rate of 0.000006.

Subsequently, simulations were conducted while varying the number of CP from three to six. In Figure 4, the steps of the calibration function from 1 to *N* can be seen. As shown in Table 4, even with only three calibration points, the relative error was 0.00381%, which is lower than the error of 0.0823% for the single M-PPC. Regarding the case in which the number of CP was three, the degree value was two, marking a significant decrease compared to the value of five in the single M-PPC. This reduction in the polynomial degree led to lower computational complexity and decreased calculation times. The segmented method showed higher accuracy at lower degrees compared to the single M-PPC method.

#### 3.1.2. One-Dimensional Calibration Simulation—A Linear Sensor

This section considers the calibration of a sensor with linear characteristics, and the segmented method is compared to the single calibration method. In the simulation, the 8-bit signed fixed-point format was used, considering the hardware constraints when applying sensor calibration in digital hardware. Many sensors in real-world applications rely on digital signal processing, particularly in fields like embedded systems, IoT devices, medical equipment, and automotive electronics, where low power consumption and miniaturization are crucial. In these environments, computational efficiency and processing speed are key, making fixed-point formats ideal for reducing computational complexity and optimizing power use. To align with these objectives, our simulation was designed to facilitate application in digital systems. The 8-bit fixed-point format enables real-time processing in digital systems, balancing precision and resource efficiency. The detailed information for each input dataset is as follows. The sensor data for the simulation were generated based on the nonlinear pressure sensor data provided by Eastsensor [29]. The measurement range of this sensor was set between 0 and 100 psi. The input function was generated to reflect the sensor’s hysteresis and nonlinear characteristics as illustrated in Figure 5, while the reference data were set to represent ideal linear behavior. CP were established at five points for each segment, and 55 data points were generated. As shown in Figure 6, the data were divided into segments. Then, the simulation was conducted by applying the respective calibration functions to each segment as shown in Figure 7.

In Figure 8, the output graphs for the single and segmented methods in one-dimensional calibration can be seen. According to the simulation results in Table 5, the error rate for a single calibration decreased by 2.70536%. In comparison, the segmented calibration method reduced the error rate by 7.49072%, demonstrating that it resulted in a lower error rate.

As shown in Table 6 and Figure 9, regarding the error rates for each segment, the segmented calibration consistently yielded lower error rates compared to the single calibration across all segments. The most significant difference was observed in the high segment, where the error rate was 9.43287% for the single calibration but was reduced to 1.25386% for the segmented calibration.

### 3.2. Two-Dimensional Calibration Simulation

Starting from the one-dimensional graph in Section 3.1, an equation related to *z* was added to expand the analysis to two dimensions, and simulations were conducted. Detailed information for each input dataset is provided below. The simulation assumed a pressure sensor influenced by the temperature. The *x*-axis measurement range for the sensor’s pressure was set from 0 to 100 psi, and the *z*-axis measurement range for the temperature was set from −20 to 80 °C. Measurement data were generated by incorporating the effect of the temperature into the input data shown in Figure 5. As with the one-dimensional simulation, the reference data were set to represent ideal linear behavior. A total of 55 data points were generated for the pressure, with an additional 21 data points for the temperature. The input data used in the simulation are shown in Figure 10. The number of calibration points (CPs) for the pressure was set at five points per section, and the number of calibration points for the temperature was set at three.

In Figure 11, the output graphs for the single and segmented methods in two-dimensional calibration can be seen, demonstrating that the calibration process was adjusted to follow the reference trend shown in Figure 10a. This indicates that the output in Figure 11 reflects a calibration aligned with the linear trend shown in Figure 10a. According to the simulation results in Figure 12, the overall error rate was 4.09392% for the single method and 2.07902% for the segmented method, indicating that the segmented method reduced the error rate by 2.0149% compared to the single method. The results for each section were as follows. Before calibration, it could be observed that the error rates in the low and high sections were higher than that for the mid section as shown in Table 7. After applying the single PPC, it was found that the error rate in the mid section remained relatively high, influenced by the nonlinearity of the low and high sections. In the segmented method, the influence between the sections was minimized, resulting in lower error rates in all sections—low, mid, and high—thereby reducing the overall error rate as well.

## 4. Discussion and Conclusions

The results of this study demonstrate that the segmented calibration method is highly effective in addressing inaccuracies in sensor measurements. By dividing the input range into multiple sections and calculating optimized calibration functions for each segment, the accuracy is improved compared to the single calibration method.

One of the main advantages of the segmented calibration method is its ability to effectively handle nonlinear regions that arise specifically within the sensor’s operating range. By isolating these regions where the sensor behavior is inconsistent with other ranges, this method prevents errors from propagating across the sensor’s data, thereby reducing the overall error rate. In this study, compared to the original single calibration approach, segmented calibration achieved lower error rates with reduced polynomial degrees. For instance, the use of only three calibration points (CP) in segmented calibration resulted in an error rate of 0.00381%, which is lower than the 0.0823% error rate observed in single calibration using six CP. This highlights the method’s ability to increase the accuracy while reducing the computational complexity and processing time.

The reduction in the polynomial degree and complexity also enabled the method to be extended to two-dimensional calibration, accounting for cross-sensitivity factors such as environmental variables. The simulation results confirmed that the method is effective even in two-dimensional calibration scenarios, offering more accurate sensor measurements. The simulation results given in Section 3.2 demonstrate that, in 8-bit signed fixed-point operations, the error rate was reduced from 15.84% to 2.07%. When comparing the error rates across individual segments between the single and segmented calibration methods, it decreased by 2.01%, confirming the effectiveness of the segmented approach. Furthermore, when comparing the segment-specific error rates between the single and segmented calibration methods, they decreased from 4.17% to 1.59%, confirming the effectiveness of the segmented approach. Specifically, regarding the sectional error rates, that in the mid section, which displayed a smaller reduction due to the calibration function being influenced by the low and high sections, was reduced from 4.45% to 2.24%. This demonstrates that the segmented method prevents error propagation throughout the data, thereby reducing the overall rate of errors.

This method offers significant scalability and can be applied to various types of sensors. The simulations described in Section 3.1 confirm its effectiveness for sensors with both linear and nonlinear characteristics. In each simulation, boundary points were set to minimize the errors, being suitable for both linear and nonlinear sensors, resulting in segmentation. The boundary points and the number of segments can be determined based on the sensor’s characteristics, allowing this method to be applied flexibly. The advantages and disadvantages of the proposed method are presented in Table 8.

In conclusion, this study demonstrates the efficacy of the segmented calibration method in addressing nonlinearity issues in conjunction with the modified PPC technique. The segmented method effectively reduced the rate of nonlinear errors, particularly near the upper and lower limits of the sensor’s range. At the same time, the modified PPC algorithm maintained the calibration accuracy while reducing the computational complexity, surpassing the original single calibration method. This approach offers a scalable solution that is capable of calibrating various sensors and environmental variables, making it a versatile calibration method with broad applicability.

## Figures and Tables

**Figure 1 sensors-24-07058-f001:**
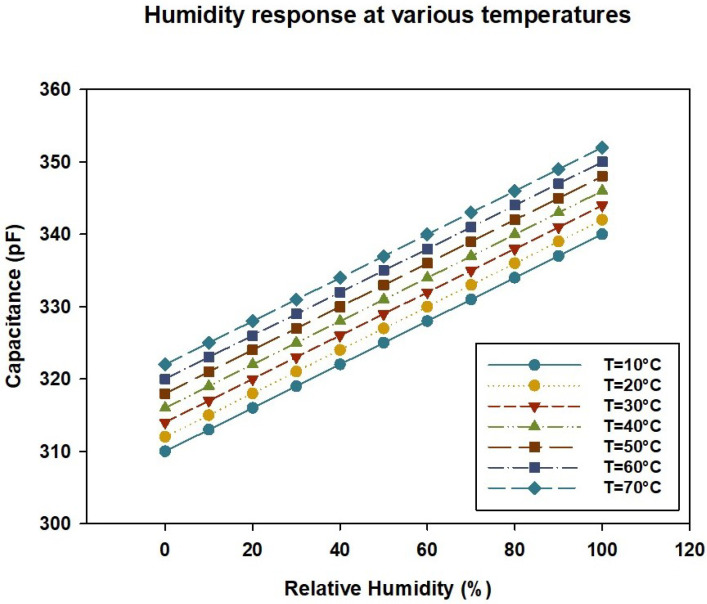
Humidity sensor’s response at various temperatures (Honeywell—HCH 1000).

**Figure 2 sensors-24-07058-f002:**
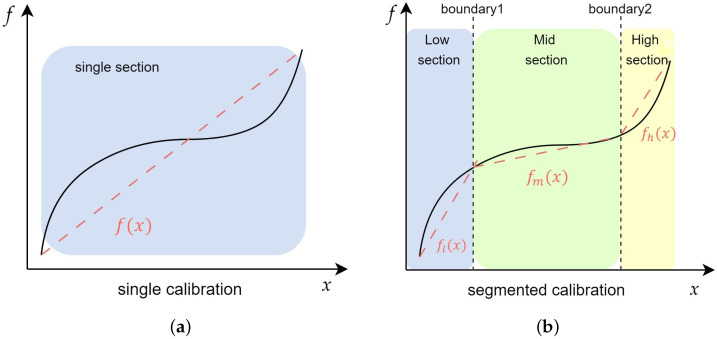
Illustration of single and segmented calibration methods. (**a**) Single calibration; (**b**) segmented calibration. The solid line represents the sensor’s input, while the dashed line indicates the calibration function calculated for each input.

**Figure 3 sensors-24-07058-f003:**
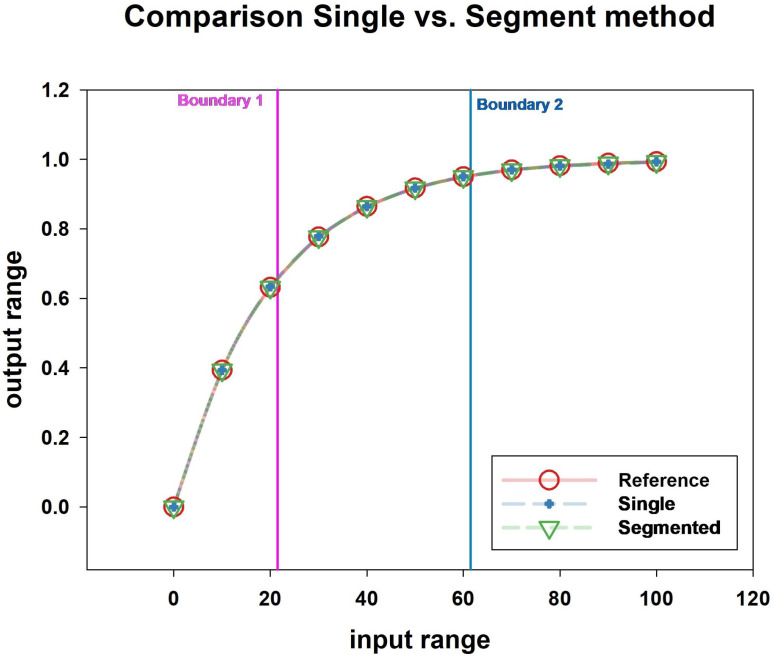
Diagram of MATLAB function operation.

**Figure 4 sensors-24-07058-f004:**
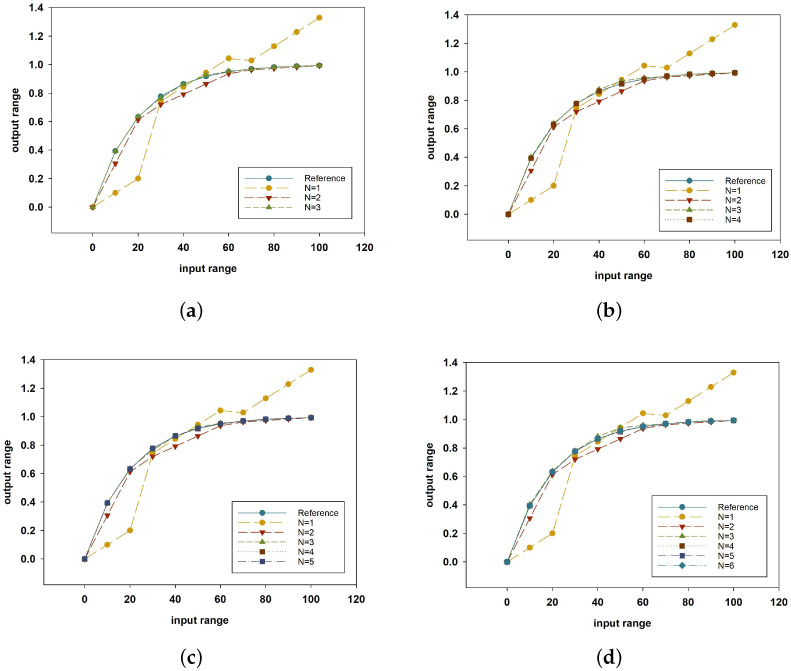
Step-by-step MATLAB simulation with different numbers of CP: (**a**) 3, (**b**) 4, (**c**) 5, and (**d**) 6.

**Figure 5 sensors-24-07058-f005:**
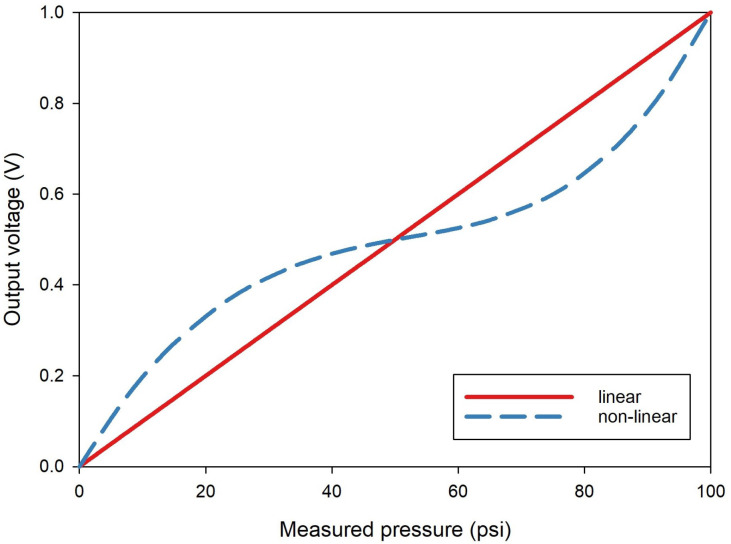
Input data.

**Figure 6 sensors-24-07058-f006:**
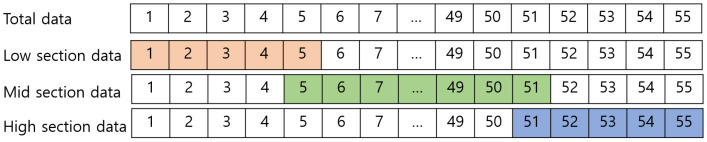
Example of segmentation based on low boundary and mid boundary.

**Figure 7 sensors-24-07058-f007:**
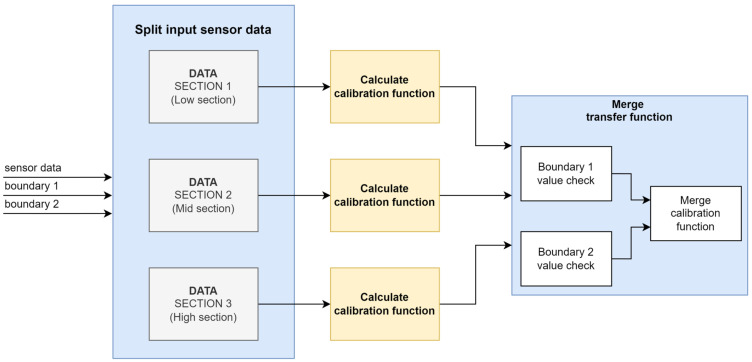
Diagram of MATLAB function operation.

**Figure 8 sensors-24-07058-f008:**
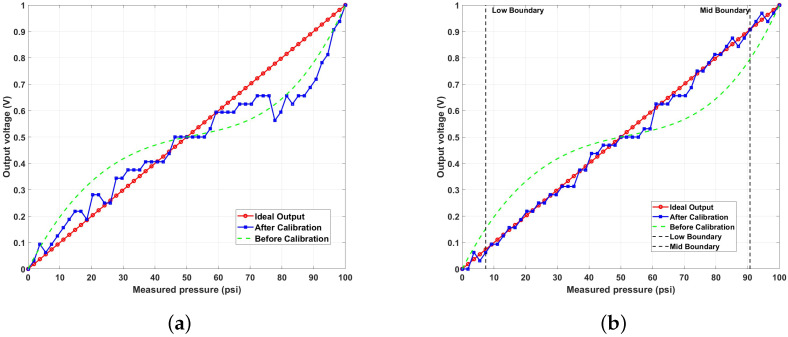
MATLAB simulation results: transfer function graph for (**a**) single calibration and (**b**) segmented calibration.

**Figure 9 sensors-24-07058-f009:**
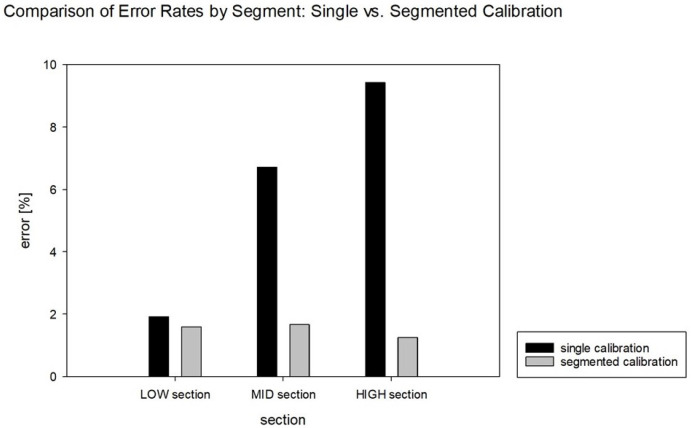
Error rate graph by segment: 1D calibration.

**Figure 10 sensors-24-07058-f010:**
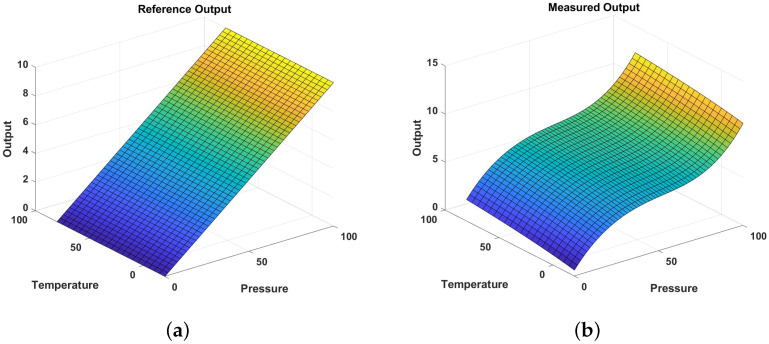
Input data graph: (**a**) reference data; (**b**) measured data. The color gradient in each plot represents variations in output values, with colors indicating the relative magnitude of data points across the surface.

**Figure 11 sensors-24-07058-f011:**
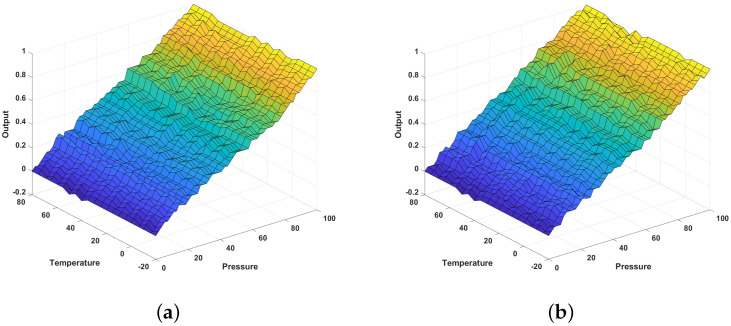
Calibration output graph: (**a**) single calibration; (**b**) segmented calibration. The color gradient in each plot represents variations in output values, with colors indicating the relative magnitude of data points across the surface.

**Figure 12 sensors-24-07058-f012:**
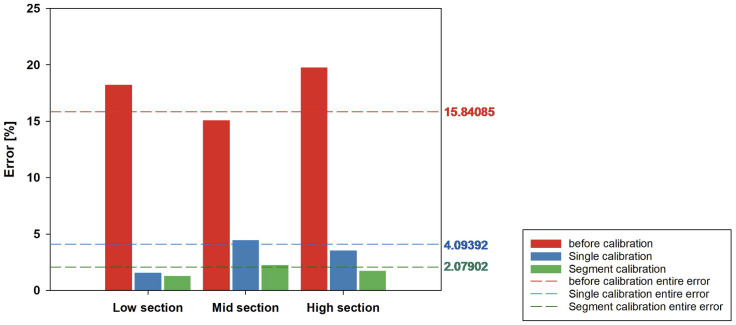
Error graph by segment: 2D calibration.

**Table 1 sensors-24-07058-t001:** Parameter descriptions: one-dimensional calibration function.

Parameter Description	
$f_{n}\left(x\right)$	calibrated function at the *n*-th step
*x*	physically measured value
*y*	reference value
n	calibration point
$a_{n}$	coefficient of PPC function at the *n*-th step
$b_{n}$	coefficient of M-PPC function at the *n*-th step

**Table 2 sensors-24-07058-t002:** Parameter descriptions: two-dimensional calibration function.

Parameter Description	
$f_{nm}\left(x\right)$	calibration function at the *n* and *m* steps
x	physically measured value
z	cross-sensitivity value
y	reference value
n	calibration point of *x*
m	calibration point of *z*
$a_{nm}$	coefficient of PPC function at the *n* and *m* steps
$b_{nm}$	coefficient of M-PPC function at the *n* and *m* steps

**Table 3 sensors-24-07058-t003:** Error rates for single and segmented cases.

Method Type	Degree	Relative Error [%]
PPC (single)	16	1.66035
M-PPC (single)	5	0.0823
M-PPC (segmented)	5	0.000006

**Table 4 sensors-24-07058-t004:** Error rates and degrees at different numbers of CP.

Number of CP	Degree	Relative Error [%]
N = 3	2	0.00381
N = 4	3	0.00057
N = 5	4	0.000031
N = 6	5	0.000006

**Table 5 sensors-24-07058-t005:** Error rates for single and segmented cases.

Before Calibration [%]	Single Calibration [%]	Segmented Calibration [%]
9.15107	6.44571	1.66035

**Table 6 sensors-24-07058-t006:** Comparison of single and segmented cases’ error rates in different sections: one dimensional.

Section	Single [%]	Segmented [%]
Low section	1.9213	1.5972
Mid section	6.7205	1.6757
High section	9.4329	1.2539

**Table 7 sensors-24-07058-t007:** Comparison of single and segmented cases’ error rates in different sections: two-dimensional.

Method	Low Section [%]	Mid Section [%]	High Section [%]
Before calibration	18.2174	15.0684	19.7522
Single calibration	1.5476	4.4531	3.5177
Segmented calibration	1.2721	2.2328	1.7072

**Table 8 sensors-24-07058-t008:** Advantages and drawbacks of proposed segmented calibration method.

Advantages	Drawbacks
Improved accuracy, especially in nonlinear regions (boundary areas)	Increased complexity from segment boundaries
Low polynomial degree and prevention of Runge phenomenon	Unnecessary processes in consistent nonlinearity
Flexible and scalable	
Minimal error propagation	

## Data Availability

The data are contained within the article.

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
