# Peer review of "Segmented Two-Dimensional Progressive Polynomial Calibration Method for Nonlinear Sensors"

_sensors, 2024, doi:10.3390/s24217058_

Round 1
Reviewer 1 Report
Comments and Suggestions for Authors
This study proposes a segmented calibration method that divides the input range into multiple sections and calculates optimized calibration functions for each. This approach reduces overall error and improves calibration accuracy by isolating distinctive regions. There are three contributions in this work. First is the segmented calibration method can effectively handle nonlinear regions; second is this method may be extended to two-dimensional calibration; the last is this method offers significant scalability and can be applied to various types of sensors. Totally speaking, this work is valuable but at the same time, the paper need to be further revised, according to the following comments.
1. The title is questionable, now that the first sentence in abstract states “Nonlinearity in sensor measurements reduces the accuracy of sensors”, according to the reviewer’s understanding, the so-called nonlinearity should be avoided and reduced in sensors as far as possible, why here “for Enhancing non-linearity in Sensor”? In addition, there are two different spelling for the same word, “nonlinearity” and “non-linearity”, please unify them; and there is an issue concerning the letter capitalization problem for word “non-linearity” in title, please check.
2. In the abstract, the authors refer to the following data “a significantly lower error rate of 0.0006%, compared to 0.0823%.”, and “a reduction in error rate from 15.84% to 2.07%”. Do these data only correspond to the input-output relation from Equation (11)? If yes, the premise for obtaining these data must be stated clearly in the abstract.
3. More importantly, the author should use an input-output characteristic from a specific sensor with application, to illustrate the advantages of the method presented in this paper, rather than the defined function like Equation (11).
4. At lines 23-24, “The effect of temperature on the output of the relative humidity sensor can be seen in Figure 1.” However, the vertical coordinate axis is capacitance, please give more explanation for Figure 1.
5. Generally, the introduction section is slightly weak (only three paragraphs with a figure), and the description of the research background and specific problem proposed should be further strengthened, and the references should be moderately increased.
6. At line 155, “… error rate of 0.0006.” while at line 9, “…error rate of 0.0006%,” Is the error rate that the authors refer to 0.0006 or 0.0006%? please check and unify it.
7. At line 72, there is a typo, please check.
8. Figure 11 is mentioned in advance at line 157, but no explanation is found below. What is the difference between (a) Single calibration and (b) Segmented calibration? The reviewers didn't see any meaningful difference. The author should give description and explanation in details.
9. At line 48, the title style of Section 2.1 is obviously different from that of next Section 2.2, please check.
10. Reference [12] is problematic, please check. In addition, the journal names should be abbreviated, and all authors should be listed (Reference [19]), please follow reference style for the journal Sensors.
Reviewer 2 Report
Comments and Suggestions for Authors
This article proposes a method that divides the sensor’s range into segments and applies an optimized calibration function to each segment. This. However, I have a few comments. Please refer to them.
1. How were the calibration points chosen for each section, and could the choice of these points introduce bias in the results?
2. What criteria were used to determine the number of segments in the segmented calibration method, and how does this affect the overall accuracy?
3. Were any limitations considered in the simulation setup that could affect the generalization of these results to real-world applications?
4. How does the segmented calibration method perform when sensor measurements have extreme nonlinearity, beyond what was simulated?
5. Can the segmented method handle dynamic sensor behavior over time, or is it only effective for static calibration?
6. Is there any risk that the segmented calibration method could introduce discontinuities or inconsistencies at the segment boundaries?
7. What was the rationale for using the 8-bit signed fixed-point format in the simulation, and how would different bit resolutions affect the error rates?
8. Was any cross-validation or external validation used to ensure the results are robust and not overfitted to the simulated data?
9. How does the proposed method compare to other existing calibration techniques regarding computational efficiency and scalability for real-time applications?
10. Could the segmented calibration method introduce additional computational complexity when applied to higher-dimensional sensor systems or more complex environments?
11. What are the remaining probable error causes in the suggested method? The authors should also include a discussion about potential limitations. Please elaborate.
12. A table summarizing the advantages and drawbacks of the suggested method would be good, as well as a quantitative comparison of the proposal to others found in the literature.
13. The article needs a fundamental review. The abstract still needs to be considered. It is too general in its present form. The abstract and conclusion sections should be rewritten. In these two sections, the innovation of the method must be clearly stated. In the introduction section, similar and more up-to-date articles should be examined.
14. In the introductory part, it is important to discuss how the remainder of the writing will be organized.
15. Please summarize the novel contributions of the current work, preferably point-wise.
Comments on the Quality of English LanguageThe English could be improved to more clearly express the research.
Round 2
Reviewer 1 Report
Comments and Suggestions for Authors
This paper may be accepted for publication in present form, but there are still two minor issues that need to be addressed. First, Figure 11 has not been mensioned in the main text; Second, the journal names should be abbreviated in References.
Reviewer 2 Report
Comments and Suggestions for Authors
I have no further comments
